# Label Text-aided Hierarchical Semantics Mining for Panoramic Activity Recognition

## ABSTRACT

Panoramic activity recognition is a comprehensive yet challenging task in crowd scene understanding, which aims to concurrently identify multi-grained human behaviors, including individual actions, social group activities, and global activities. Previous studies tend to capture cross-granularity activity-semantics relations from solely the video input, thus ignoring the intrinsic semantic hierarchy in label-text space. To this end, we propose a label text-aided hierarchical semantics mining (THSM) framework, which explores multi-level cross-modal associations by learning hierarchical semantic alignment between visual content and label texts. Specifically, a hierarchical encoder is first constructed to encode the visual and text inputs into semantics-aligned representations at different granularities. To fully exploit the cross-modal semantic correspondence learned by the encoder, a hierarchical decoder is further developed, which progressively integrates the lower-level representations with the higher-level contextual knowledge for coarse-to-fine action/activity recognition. Extensive experimental results on the public JRDB-PAR benchmark validate the superiority of the proposed THSM framework over state-of-the-art methods.

## CCS CONCEPTS

• **Computing methodologies → Activity recognition and understanding**.

## KEYWORDS

Panoramic Activity Recognition, Hierarchical Semantics Mining, Vision-Language Learning

## 1 INTRODUCTION

Human activity recognition (HAR), which aims to automatically interpret or identify behaviors occurring in scenes, has attracted considerable attention in both academic and industrial communities, owing to its widespread real-world applications [18, 26, 31, 46], such as intelligent surveillance, social events analysis, and multimedia content review. Over the past decade, researchers have made various attempts to recognize activities at a specific granularity level, e.g., single subject-based individual actions [3, 39], a few people-involved interaction activities [28, 29], and group activities in crowd scenes [33, 44]. However, in practical unconstrained environments, it is likely that the scenes contain multi-grained semantic levels of activities, which pose great challenges for the existing HAR methods. Thus, in this paper, we focus on addressing an emerging activity understanding task, namely panoramic activity recognition (PAR), which requires models to comprehensively recognize behaviors in crowded scenes from three semantic granularities, including individual action, social group activity, and global (scene) activity. This is essentially a challenging task, with the need to establish latent relationships among the human activities of different granularity levels.

Previous PAR works exploited the hierarchical graph network [12] or the Transformer-based perception block [2] to explore cross-granularity activity-semantics relations from the sampled visual input. However, they neglect the inherent semantic hierarchy in the label-text space, which can be resorted to build rich cross-modal correspondence at multiple levels. For instance, the semantic relation between the visual content of one of the subjects in a group and the corresponding individual-action text of "listening to someone" or "talking to someone" is crucial in inferring the social group activity of "chatting". Moreover, identifying the correspondence between the holistic scene and the corresponding global-activity text of "walking" first can provide prior knowledge regarding the overall event type, which eases the reasoning of social group activities, e.g., "walking closely", and is beneficial to suppress unreasonable predictions, e.g., "sitting closely".

More generally, as illustrated in Fig. 1, there are intrinsically two flows of semantic hierarchies in tackling the PAR task. On the text side (see Fig. 1(a)), the semantics of the label set can be naturally divided into a three-level hierarchy, which consists of individual action, social group activity and global activity organized in a bottom-up manner. On the visual side (see Fig. 1(b)), the appearance clues corresponding to different semantic levels of activities also exhibit three granularities, with spatially interaction-based dependencies from coarse to fine. By learning the associations between the visual content and different levels of label texts for multi-level action/activity recognition, the model is encouraged to explore the interconnections between activities at different semantic granularities.

Based on the abovementioned observations, we propose a label text-aided hierarchical semantics mining (THSM) method for PAR. Our proposed THSM framework is devised based on a multi-level encoder-decoder architecture, which exploits hierarchical semantic clues from two perspectives.

**Unpublished working draft. Not for distribution.**

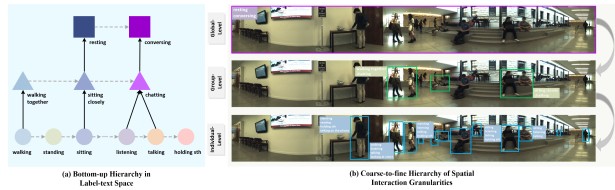

(a) Bottom-up Hierarchy in Label-text Space

(b) Coarse-to-fine Hierarchy of Spatial Interaction Granularities

**Figure 1: Illustration of (a) the bottom-up semantic hierarchy of label-text space corresponding to individual action, social group activity, and global activity, and (b) the coarse-to-fine hierarchy of spatial interaction granularities exhibited in visual clues.**

On the one hand, the hierarchical cross-modal correspondence is learned by gradually aligning the semantics between visual input and label texts in multiple levels of common spaces. Precisely, to fully explore the semantic hierarchy of label texts, we construct a hierarchical cross-modal encoder, which comprises three semantic granularities, including individual, social group, and global scene, from the bottom up. Each level of the encoder receives the action/activity category text embeddings and the pooled visual representations from a lower level as input, and continually learns visual-textual associations at a higher level via attention-based cross-modal interactions. On the other hand, we leverage the learned abundant cross-level contexts to progressively perform multiple levels of action/activity recognition in a coarse-to-fine fashion. Concretely, we design a three-level coarse-to-fine decoder based on the spatial interaction granularities from global to local. Each level of the decoder progressively integrates the lower-level cross-modal representations with the higher-level contextual knowledge, which facilitates finer action/activity reasoning with the guidance of holistic event semantics. Thus, the three sub-tasks in PAR are jointly conducted within the proposed unified hierarchical framework, which is beneficial to transfer useful clues across different levels.

The main contributions of this paper can be summarized in three ways. 1) A label text-aided hierarchical semantics mining (THSM) framework is proposed for panoramic activity recognition. To the best of our knowledge, this is the first work that explores hierarchical cross-modal semantic correspondence between the visual content and label texts for improving PAR. 2) A multi-level encoder-decoder architecture is designed, where the encoder accounts for visual-textual semantics alignment at different granularities, while the decoder progressively integrates the learned cross-level cross-modal semantics for coarse-to-fine action/activity recognition. 3) Extensive experimental results and ablation studies on the public JRDB-PAR benchmark validate that the proposed THSM framework can consistently outperform other competing methods.

## 2 RELATED WORK

**Human Activity Recognition.** As one of the longstanding research topics, human activity recognition (HAR) has

gained great improvements with the rapid development of deep learning techniques. 1) 3-D CNNs-based methods [3, 35] simultaneously learn spatial and temporal features via stacked 3-D convolution and pooling operations. To alleviate the high computational cost brought by 3-D CNNs, several attempts have been made to replace a 3-D convolution kernel with a 2-D spatial kernel and a 1-D temporal kernel, e.g., R(2+1)D [36] and S3D [42]. 2) The two-stream CNN architecture [30] receives RGB and optical-flow inputs to separately extract appearance and motion representations for activity recognition. Then, several methods employed the basic idea of the two-stream architecture to design a multi-stream network for learning diverse features, with efficient motion-aware blocks, e.g., STM [15], or with different RGB sequences sampled at various frame rates, e.g., SlowFastNet [8]. 3) Transformer-based HAR methods, e.g., TimeSformer [1], VideoSwin [22], and DVT [38], are typically built based on the ViT [4] model, by regarding the time axis as an extra dimension and formulating diverse temporal attention mechanisms to measure the similarities among patches in different frames. To reduce the computational cost of video Transformers, MViT [7] employs a series of local pooling operations, which gradually reduce the number of tokens while increasing the channel dimension.

**Activity Understanding in Multi-person Scenes.** As a pioneering task for understanding activities in multi-person scenes, group activity recognition (GAR) targets at identifying activities performed by a group of individuals. Over the past few years, deep learning-based methods have achieved promising performance on GAR. Ibrahim et al. [14] first designed a two-stage deep model with two LSTM modules, which extracts the individual-level action dynamics and learns group-level representations, respectively. Since multiple persons in the scenes can be naturally modeled by attributed graphs, where the individuals and interactions correspond to the nodes and the edges, respectively, graph neural network (GNN) has been employed for tackling the GAR task [5]. Wu et al. [41] proposed an actor relation graph (ARG) by measuring both the appearance and position relationship between subjects, and utilized a graph convolutional network (GCN) for inferring group activities. Xie et al. [43] proposed an actor-centric causality graph to model the asynchronous temporal causal relationships among individuals in the scenes. Inspired by the excellent capacity of Transformers [37] in capturing long-term dependencies [11], Li et al. [20] devised a clustered spatial-temporal transformer to enhance the individual and group features, by concurrently capturing spatial and temporal contexts. Recently, to comprehensively understand multi-granularity activities occurring in the crowd scenes, Han et al. [12] introduced a new task, namely panoramic activity recognition (PAR), and developed a hierarchical graph network to progressively recognize the activities from different semantic levels. Cao et al. [2] proposed a unified perception framework based on Transformer blocks, to synchronously excavate both intra- and cross-granularity semantics for PAR. Different from these methods, we resort

to the aid of the unleashed semantic hierarchy in the label-text space, which can be leveraged to establish cross-modal correspondence, thereby facilitating activity recognition at multiple granularities.

**Hierarchical Vision-Language Learning.** In the past few years, with the remarkable progress of vision-language pretraining (VLP), learning hierarchical vision-language representations from image-text pairs has attracted increasing attention and benefited diverse downstream tasks [16, 19, 27, 34, 48]. PyramidCLIP [9] first builds a pyramid with different semantic levels for each input modality, and aligns visual and linguistic entities by exploiting both peer-level semantics and cross-level relations. MVPTR [21] divides hierarchical multi-modal alignment learning into two phases, which conduct intra-modality multi-level representation learning and cross-modal interactions, respectively. X-VLM [50] learns multi-grained alignments between the discovered visual concepts in the image and the associated texts. Motivated by the effectiveness of the aforementioned works, the proposed THSM method explores hierarchical semantic associations between visual content and label texts for improving multi-level action/activity understanding in crowd scenes.

## 3  METHODOLOGY

As illustrated in Fig. 2, the overall framework is designed based on a hierarchical encoder-decoder architecture. First, we encode the sampled input frames and three-level label texts (including individual action, social group activity, and global activity) into embeddings. Then, these visual and text embeddings are fed into a hierarchical encoder to learn representations at different semantic granularities. By fully exploring the visual-textual associations, a hierarchical decoder is further leveraged to progressively integrate the learned cross-modal semantics for coarse-to-fine action/activity recognition in the crowd scenes. In this way, the three sub-tasks are jointly conducted through the hierarchical framework, which facilitates the sharing of beneficial knowledge for panoramic activity understanding.

### 3.1  Cross-Modal Embedding Extraction

**Visual Embedding.** Given a video captured from crowd scenes, we sample several frames as the input and employ a pretrained CNN network, e.g., Inception-v3 [32], to extract initial visual feature maps. Then, based on the bounding box of each person, the local features of each individual are cropped from these feature maps and normalized to the same size via RoIAlign [13]. The individual-level feature of the $i$-th person is denoted as $\mathbf{f}_i \in \mathbb{R}^{H \times W \times C}$, where $C$ is the number of channels, and $H$ and $W$ are the height and width of the local feature map, respectively. By following ViT [4], we further flatten the individual feature $\mathbf{f}_i$ into a series of patches, denoted as $\mathbf{f}_i^p \in \mathbb{R}^{N \times (P^2 \times C)}$, where $(P, P)$ represents the resolution of each patch, and $N = HW/P^2$ is the number of patches. A trainable linear layer is employed to project the patches into D-dimensional visual embeddings $\mathbf{f}_i^v \in \mathbb{R}^{N \times D}$.

**Text Embedding.** Given the label-text set of three-level action/activity granularities, we first leverage a pretrained Glove [24] model to convert each category text into a vector embedding. Then, a three-layer self-attention module is employed to generate label-text embeddings as $\mathbf{F}^{\mathcal{I}} = \left\{ \mathbf{f}_j^{\mathcal{I}} \right\}_{j=1}^{L^{\mathcal{I}}} \in \mathbb{R}^{L^{\mathcal{I}} \times D}$, $\mathbf{F}^{\mathcal{S}} = \left\{ \mathbf{f}_j^{\mathcal{S}} \right\}_{j=1}^{L^{\mathcal{S}}} \in \mathbb{R}^{L^{\mathcal{S}} \times D}$, and $\mathbf{F}^{\mathcal{G}} = \left\{ \mathbf{f}_j^{\mathcal{G}} \right\}_{j=1}^{L^{\mathcal{G}}} \in \mathbb{R}^{L^{\mathcal{G}} \times D}$, where $L^{\mathcal{I}}$, $L^{\mathcal{S}}$, and $L^{\mathcal{G}}$ denote the categories of individual action, social group activity, and global activity, respectively.

### 3.2  Hierarchical Cross-Modal Encoder

**Individual-level Encoder.** Given the visual embeddings $\mathbf{F}^{\mathcal{V}} = \{\mathbf{f}_i^v\}_{i=1}^M \in \mathbb{R}^{M \times N \times D}$, where $M$ denotes the number of individuals in the sampled frame, we first utilize a self-attention module to produce refined patch-level embeddings $\mathbf{F}^{\mathcal{E}} = \{\mathbf{f}_i^e\}_{i=1}^M$. Then, a patch pooling operation is employed to obtain the global tokens of individuals, as follows:

$$\mathbf{\Omega}_i^{\mathcal{I}} = \frac{1}{N} \sum_{n=1}^N \mathbf{f}_i^e (n), \tag{1}$$

where $n$ represents the patch index within the local regions of each individual. We further exploit the cross-attention mechanism [37], which utilizes the global tokens as queries to induce the initial individual-level visual representation $\mathbf{X}^{\mathcal{I}(0)}$, as follows:

$$\mathbf{X}^{\mathcal{I}(0)} = \text{CrossAttn}\left(\mathbf{\Omega}^{\mathcal{I}}, \mathbf{F}^{\mathcal{E}}, \mathbf{F}^{\mathcal{E}}\right). \tag{2}$$

Subsequently, an individual-level encoder is devised to learn the shallow cross-modal semantic relations between the initial visual input $\mathbf{X}^{\mathcal{I}(0)} \in \mathbb{R}^{M \times D}$ and action-label texts $\mathbf{Q}^{\mathcal{I}(0)} \in \mathbb{R}^{L^{\mathcal{I}} \times D}$ ($\mathbf{Q}^{\mathcal{I}(0)} = \mathbf{F}^{\mathcal{I}}$). Specifically, in the $(k + 1)$-st layer, we first concatenate the cross-modal inputs and project them into a shared representation $\mathbf{U}^{\mathcal{I}(k)} \in \mathbb{R}^{(M+L^{\mathcal{I}}) \times D}$. Then, a cross-modal self-attention mechanism is employed to measure pairwise semantic affinities, as follows:

$$\mathbf{s}_{ij}^{\mathcal{I}(k)} = \frac{1}{\sigma^k} \cdot \frac{\varphi^{(k)}\left(\mathbf{U}_i^{\mathcal{I}(k)}\right)\left(\phi^{(k)}\left(\mathbf{U}_j^{\mathcal{I}(k)}\right)\right)^{\mathrm{T}}}{\left\|\varphi^{(k)}\left(\mathbf{U}_i^{\mathcal{I}(k)}\right)\right\|\left\|\phi^{(k)}\left(\mathbf{U}_j^{\mathcal{I}(k)}\right)\right\|}, \tag{3}$$

where $\varphi^{(k)}(\cdot)$ and $\phi^{(k)}(\cdot)$ are two learnable linear projection functions, the scalar coefficient $\sigma^{(k)}$ controls the sharpness of the similarity function, and $\|\cdot\|$ denotes $L_2$ norm. Thereafter, the similarity scores $\mathbf{s}^{\mathcal{I}(k)}$ are utilized to rearrange and aggregate the cross-modal semantics, as follows:

$$\mathbf{H}^{\mathcal{I}(k+1)} = \text{softmax}\left(\mathbf{s}^{\mathcal{I}(k)}\right)\psi^{(k)}\left(\mathbf{U}^{\mathcal{I}(k)}\right), \tag{4}$$

where $\psi^{(k)}(\cdot)$ is another trainable linear projection function. After the cross-modal self-attention layer, two modality-specific multilayer perceptron (MLPs) are introduced to derive visual output $\mathbf{X}^{\mathcal{I}(k+1)}$ and textual output $\mathbf{Q}^{\mathcal{I}(k+1)}$ from the cross-modal representation $\mathbf{H}^{\mathcal{I}(k+1)}$. To ensure sufficient alignment learning between the visual content of each individual and the action-label texts, we stack $B_1$ layers to perform the complicated cross-modal interactions, i.e., $k \in$

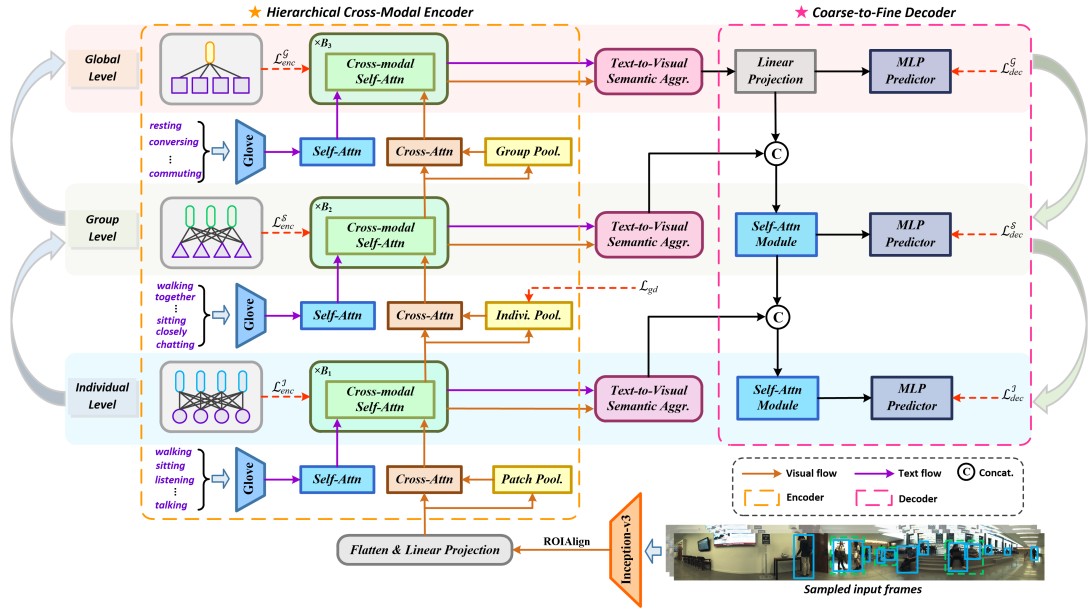

**Figure 2: The overall architecture of the proposed label text-aided hierarchical semantics mining (THSM) framework, which consists of a hierarchical cross-modal encoder and a coarse-to-fine decoder.**

$\{0, 1, ..., B_1 - 1\}$. Thus, the individual-level encoder captures the low-level visual-textual semantic associations, which serve as the basis for the understanding of higher-level activity concepts.

**Group-level Encoder.** Based on the social group division results, we first employ an individual pooling operation and a cross-attention operation on $\mathbf{X}^{\mathcal{I}(B_1)}$, to obtain the initial group-level visual representation $\mathbf{X}^{\mathcal{S}(0)} \in \mathbb{R}^{S \times D}$, where $S$ denotes the number of social groups detected in the scene. Then, a cross-modal representation $\mathbf{U}^{\mathcal{S}(0)} \in \mathbb{R}^{(S+L^{\mathcal{S}}) \times D}$ can be generated by concatenating and projecting the group-level visual input $\mathbf{X}^{\mathcal{S}(0)}$ and activity-label texts $\mathbf{Q}^{\mathcal{S}(0)} \in \mathbb{R}^{L^{\mathcal{S}} \times D}$ ($\mathbf{Q}^{\mathcal{S}(0)} = \mathbf{F}^{\mathcal{S}}$). A series of stacked cross-modal self-attention layers and two unshared MLPs are utilized to produce the visual and textual output $\mathbf{X}^{\mathcal{S}(B_2)}$ and $\mathbf{Q}^{\mathcal{S}(B_2)}$, where $B_2$ represents the number of group-level layers. The group-level encoder builds the visual-textual correspondence at the middle semantic granularity, which bridges the atomic individual-level actions and holistic scene-level events.

**Global-level Encoder.** Analogously, a group pooling operation followed by a cross-attention module is applied to $\mathbf{X}^{\mathcal{S}(B_2)}$ to obtain an initial global-level visual representation $\mathbf{X}^{\mathcal{G}(0)} \in \mathbb{R}^D$. After constructing the cross-modal representation $\mathbf{U}^{\mathcal{G}(0)} \in \mathbb{R}^{(L^{\mathcal{G}}+1) \times D}$ by concatenation and projection of the group-level visual and label-text input $\mathbf{X}^{\mathcal{G}(0)}$ and $\mathbf{Q}^{\mathcal{G}(0)} \in \mathbb{R}^{L^{\mathcal{G}} \times D}$ ($\mathbf{Q}^{\mathcal{G}(0)} = \mathbf{F}^{\mathcal{G}}$), a stack of $B_3$ cross-modal self-attention layers and two modality-specific MLPs are employed on $\mathbf{U}^{\mathcal{G}(0)}$ to generate the visual output $\mathbf{X}^{\mathcal{G}(B_3)}$ and textual output $\mathbf{Q}^{\mathcal{G}(B_3)}$. The global-level encoder establishes the correspondence between the visual content of the holistic scene and the most abstract semantics of the crowd event.

**Text-to-Visual Semantic Aggregation.** To further exploit the cross-modal associations learned by the hierarchical encoder for panoramic activity recognition, we aggregate the label-text clues into visual representations according to the visual-to-textual semantic affinities, as follows:

$$\mathbf{A}^{\ell} = \frac{1}{\tau^{\ell}} \cdot \overline{\mathbf{X}}^{\ell(\mathcal{D})} \left( \overline{\mathbf{Q}}^{\ell(\mathcal{D})} \right)^{\mathrm{T}}, \tag{5}$$

$$\mathbf{g}^{\ell} = \left[ \overline{\mathbf{X}}^{\ell(\mathcal{D})}; \mathrm{softmax} \left( \mathbf{A}^{\ell} \right) \overline{\mathbf{Q}}^{\ell(\mathcal{D})} \right], \tag{6}$$

where $(\ell, \mathcal{D}) \in \{ (\mathcal{I}, B_1), (\mathcal{S}, B_2), (\mathcal{G}, B_3) \}$, i.e., $\ell$ and $\mathcal{D}$ represent the semantic granularity level and the depth of each encoder, respectively. $\overline{\mathbf{X}}^{\ell(\mathcal{D})}$ and $\overline{\mathbf{Q}}^{\ell(\mathcal{D})}$ are obtained by applying $L_2$ normalization to $\mathbf{X}^{\ell(\mathcal{D})}$ and $\mathbf{Q}^{\ell(\mathcal{D})}$, respectively. $\tau^{\ell}$ is a temperature factor and [;] denotes the channel-wise concatenation. Hence, $\mathbf{A}^{\mathcal{I}}$, $\mathbf{A}^{\mathcal{S}}$, and $\mathbf{A}^{\mathcal{G}}$ reflect the cross-modal semantic similarity degrees at individual-level, group-level, and global-level, respectively. The cross-modal representation $\mathbf{g}^{\ell}$ integrates the refined visual features and the relevant semantic clues conveyed by label texts at granularity level $\ell$.

## 3.3 Coarse-to-Fine Decoder

**Global-level Decoder.** For predicting the global activity at the coarsest semantic level, we directly apply a linear projection followed by a two-layer MLP predictor on $\mathbf{g}^{\mathcal{G}}$, to obtain the activity classification results $\widetilde{\mathbf{y}}^{\mathcal{G}} \in \mathbb{R}^{L^{\mathcal{G}}}$ of the holistic scene.

**Group-level Decoder.** At the group level, the goal is to identify the interactive activities occurring in each detected social group. Moreover, contextual cues from the global-level decoder can provide holistic semantics of the crowd scene. Thus, we integrate the group-level and global-level

cross-modal representations $\mathbf{g}^{\mathcal{S}}$ and $\mathbf{g}^{\mathcal{G}}$, and feed them into a self-attention module, as follows:

$$\left\{ \mathbf{g}_r^{\mathcal{SG}} \right\}_{r=1}^{S} = \mathrm{SelfAttn}\left( \left\{ \left[ \mathbf{g}_r^{\mathcal{S}}; \mathbf{g}^{\mathcal{G}} \right] \right\}_{r=1}^{S} \right), \qquad (7)$$

where $\mathbf{g}^{\mathcal{S}}$ functions as the conditional context that boosts the social group-level activity recognition. Then, we input $\mathbf{g}^{\mathcal{SG}}$ into a two-layer MLP predictor to obtain the activity categories for each group, i.e., $\left\{ \widetilde{\mathbf{y}}_r^{\mathcal{S}} \right\}_{r=1}^{S} \in \mathbb{R}^{S \times L^{\mathcal{S}}}$.

**Individual-level Decoder.** For the finest-level decoding, we conduct atomic action recognition with respect to each individual in the scene. Concretely, we first rearrange the original individual-level tokens $\mathbf{g}^{\mathcal{I}} \in \mathbb{R}^{M \times D}$ into the format distributed in each group, denoted as $\widehat{\mathbf{g}}^{\mathcal{I}} \in \mathbb{R}^{S \times O \times D}$, where $O$ is the padding size that is set to the maximum number of individuals in the group. Then, we feed $\widehat{\mathbf{g}}^{\mathcal{I}}$ and $\mathbf{g}^{\mathcal{SG}}$ into a self-attention module, to learn the individual-level interaction contexts, as follows:

$$\left\{ \widehat{\mathbf{g}}_{i,j}^{\mathcal{IS}} \right\}_{j=1}^{M_i} = \mathrm{SelfAttn}\left( \left\{ \left[ \widehat{\mathbf{g}}_{i,j}^{\mathcal{I}}; \mathbf{g}_i^{\mathcal{SG}} \right] \right\}_{j=1}^{M_i} \right), \qquad (8)$$

where $\widehat{\mathbf{g}}^{\mathcal{IS}}$ is the refined individual-level representation augmented with the context of the group to which it belongs. A two-layer MLP is employed on $\widehat{\mathbf{g}}^{\mathcal{IS}}$ to predict individual-level action recognition results $\widehat{\mathbf{y}}^{\mathcal{I}} \in \mathbb{R}^{S \times O \times L^{\mathcal{I}}}$. By removing the padded individuals, we can further obtain the final individual-level predictions $\left\{ \widetilde{\mathbf{y}}_i^{\mathcal{I}} \right\}_{i=1}^{M} \in \mathbb{R}^{M \times L^{\mathcal{I}}}$.

## 3.4 Training Strategy

**Encoder Loss.** To guide the learning of the hierarchical cross-modal encoder, we formulate a three-level semantic alignment loss $\mathcal{L}_{enc}$. This loss intrinsically encourages to learn the associations between visual content and label texts at different semantic granularities, as follows:

$$\mathcal{L}_{enc} = \mathcal{L}_{enc}^{\mathcal{I}} + \mathcal{L}_{enc}^{\mathcal{S}} + \mathcal{L}_{enc}^{\mathcal{G}} \qquad (9)$$

$$= \sum_{i=1}^{M} \mathcal{L}_{bce}\left( \mathbf{A}_i^{\mathcal{I}}, \mathbf{y}_i^{\mathcal{I}} \right) + \sum_{r=1}^{S} \mathcal{L}_{bce}\left( \mathbf{A}_r^{\mathcal{S}}, \mathbf{y}_r^{\mathcal{S}} \right) + \mathcal{L}_{bce}\left( \mathbf{A}^{\mathcal{G}}, \mathbf{y}^{\mathcal{G}} \right),$$

where $\mathcal{L}_{bce}$ represents the binary cross-entropy loss function. Thus, $\mathcal{L}_{enc}^{\mathcal{I}}$, $\mathcal{L}_{enc}^{\mathcal{S}}$, and $\mathcal{L}_{enc}^{\mathcal{G}}$ measure the difference between the ground-truth labels (i.e., $\mathbf{y}^{\mathcal{I}} \in \mathbb{R}^{M \times L^{\mathcal{I}}}$, $\mathbf{y}^{\mathcal{S}} \in \mathbb{R}^{S \times L^{\mathcal{S}}}$, and $\mathbf{y}^{\mathcal{G}} \in \mathbb{R}^{L^{\mathcal{G}}}$) and the visual-text semantic affinity matrices (i.e., $\mathbf{A}^{\mathcal{I}}$, $\mathbf{A}^{\mathcal{S}}$, and $\mathbf{A}^{\mathcal{G}}$ derived from Eq. (5)) at individual, group, and global levels, respectively.

**Decoder Loss.** We leverage multiple levels of classification loss on the action/activity category prediction results produced by the hierarchical coarse-to-fine decoder, as follows:

$$\mathcal{L}_{dec} = \mathcal{L}_{dec}^{\mathcal{I}} + \mathcal{L}_{dec}^{\mathcal{S}} + \mathcal{L}_{dec}^{\mathcal{G}} \qquad (10)$$

$$= \sum_{i=1}^{M} \mathcal{L}_{bce}\left( \widetilde{\mathbf{y}}_i^{\mathcal{I}}, \mathbf{y}_i^{\mathcal{I}} \right) + \sum_{r=1}^{S} \mathcal{L}_{bce}\left( \widetilde{\mathbf{y}}_r^{\mathcal{S}}, \mathbf{y}_r^{\mathcal{S}} \right) + \mathcal{L}_{bce}\left( \widetilde{\mathbf{y}}^{\mathcal{G}}, \mathbf{y}^{\mathcal{G}} \right),$$

where $\mathcal{L}_{dec}^{\mathcal{I}}$, $\mathcal{L}_{dec}^{\mathcal{S}}$, and $\mathcal{L}_{dec}^{\mathcal{G}}$ represent the classification losses with respect to individual action, social group activity, and global activity, respectively.

**Group Detection Loss.** Panoramic activity recognition involves the subtask of social group detection. This subtask aims to discover the group that has relatively strong interactions between individuals. Thus, the results of group division are crucial for the individual-to-group representation pooling in the group-level encoder. Following previous works [2, 12], we adopt a group detection loss as follows:

$$\mathcal{L}_{gd} = \mathcal{L}_{bce}\left( \widetilde{\mathbf{Z}}, \mathbf{Z} \right), \qquad (11)$$

where $\mathbf{Z} \in \mathbb{R}^{M \times M}$ denotes the ground-truth individual-relation matrix with binary values, whose elements equal 1 only when the corresponding two subjects belong to the same group.

Therefore, the proposed hierarchical encoder-decoder-based framework is trained by jointly minimizing the loss terms defined in Eqs. (9)-(11), as follows:

$$\mathcal{L}_{total} = \mathcal{L}_{enc} + \mathcal{L}_{dec} + \mathcal{L}_{gd}. \qquad (12)$$

## 4 EXPERIMENTS

### 4.1 Experimental Setup

**Data Sets.** The proposed method is evaluated on a recently released data set, namely JRDB-PAR [12], which is tailored for panoramic activity recognition. It contains $360°$ RGB videos captured by a mobile robot in diverse crowded multi-person scenes, e.g., campuses, canteens, and classrooms, etc. The JRDB-PAR benchmark inherits the annotations of human bounding boxes with IDs, individual actions, and group divisions from previous the JRDB [23] and JRDB-Act [6] data sets. Additionally, it introduces manual labels for social group activities and global activities. JRDB-PAR contains 27 videos, which are further split into 20 videos for training and 7 videos for testing. In total, JRDB-PAR consists of 27,920 frames with more than 628k bounding boxes, and covers 27 categories of individual actions, 11 categories of social group activities, and 7 categories of global activities.

**Evaluation Metrics.** Following the pioneering work [12], the commonly used precision, recall and F1 score are adopted as the main evaluation metrics. For individual action recognition, the precision, recall, and F1 score are denoted as $\mathcal{P}_i$, $\mathcal{R}_i$, and $\mathcal{F}_i$, respectively, which evaluate the action classification accuracy for each subject in the testing set. For social group detection, we follow the general protocol in [40]. Moreover, after group division, we compute the precision $\mathcal{P}_p$, recall $\mathcal{R}_p$, and F1 score $\mathcal{F}_p$ as the evaluation metrics for social activity recognition. For global activity recognition, we also adopt the precision, recall, and F1 score, denoted as $\mathcal{P}_g$, $\mathcal{R}_g$, and $\mathcal{F}_g$, respectively, for evaluation. Finally, the above three F1 scores (i.e., $\mathcal{F}_i$, $\mathcal{F}_p$, and $\mathcal{F}_g$) are averaged as the overall metric $\mathcal{F}_a$ for evaluating the performance of panoramic activity recognition, i.e., $\mathcal{F}_a = \frac{1}{3}\left( \mathcal{F}_i + \mathcal{F}_p + \mathcal{F}_g \right)$.

**Implementation Details.** We employ an Inception-v3 [32] network, pretrained on ImageNet, to extract initial visual features from each sampled frame. A pretrained Glove [24] model is utilized to extract linguistic embeddings for the action/activity label texts. The number of hierarchical encoder layers $\{B_1, B_2, B_3\}$ is set to $\{2, 2, 2\}$. The cross-modal

**Table 1: Comparison results of different methods under clustered group division setting.**

| Methods | Individual Action | | | Group Activity | | | Global Activity | | | Overall |
|---|---|---|---|---|---|---|---|---|---|---|
| | $\mathcal{P}_i$ | $\mathcal{R}_i$ | $\mathcal{F}_i$ | $\mathcal{P}_p$ | $\mathcal{R}_p$ | $\mathcal{F}_p$ | $\mathcal{P}_g$ | $\mathcal{R}_g$ | $\mathcal{F}_g$ | $\mathcal{F}_a$ |
| ARG [41] | 39.9 | 30.7 | 33.2 | 8.7 | 8.0 | 8.2 | 63.6 | 44.3 | 50.7 | 30.7 |
| SA-GAT [5] | 44.8 | 40.4 | 40.3 | 8.8 | 8.9 | 8.8 | 36.7 | 29.9 | 31.4 | 26.8 |
| JRDB-Base [6] | 19.1 | 34.4 | 23.6 | 14.3 | 12.2 | 12.8 | 44.6 | 46.8 | 45.1 | 27.2 |
| PAR [12] | 51.0 | 40.5 | 43.4 | 24.7 | 26.0 | 24.8 | 52.8 | 31.8 | 38.8 | 35.6 |
| MUP [2] | 55.4 | 44.8 | 47.7 | 25.4 | 26.6 | 25.1 | 58.0 | 49.0 | 51.8 | 41.5 |
| THSM (Ours) | **58.2** | **47.3** | **50.1** | **27.3** | **29.4** | **27.3** | **66.3** | **53.6** | **57.8** | **45.1** |

**Table 2: Comparison results of different methods under ground-truth group division setting.**

| Methods | Individual Action | | | Group Activity | | | Global Activity | | | Overall |
|---|---|---|---|---|---|---|---|---|---|---|
| | $\mathcal{P}_i$ | $\mathcal{R}_i$ | $\mathcal{F}_i$ | $\mathcal{P}_p$ | $\mathcal{R}_p$ | $\mathcal{F}_p$ | $\mathcal{P}_g$ | $\mathcal{R}_g$ | $\mathcal{F}_g$ | $\mathcal{F}_a$ |
| AT [10] | 38.9 | 33.9 | 34.6 | 32.5 | 32.3 | 32.0 | 21.2 | 19.1 | 19.8 | 28.8 |
| SACRF [25] | 31.3 | 23.6 | 25.9 | 25.7 | 24.5 | 24.8 | 42.9 | 35.5 | 37.6 | 29.5 |
| TCE+STBiP [47] | 40.7 | 33.4 | 35.1 | 33.5 | 30.1 | 30.9 | 37.5 | 27.1 | 30.6 | 32.2 |
| HiGCIN [45] | 34.6 | 26.4 | 28.6 | 34.2 | 31.8 | 32.2 | 39.3 | 30.1 | 33.1 | 31.3 |
| ARG [41] | 42.7 | 34.7 | 36.6 | 27.4 | 26.1 | 26.2 | 26.9 | 21.5 | 23.3 | 28.8 |
| SA-GAT [5] | 39.6 | 34.5 | 35.0 | 32.5 | 32.5 | 30.7 | 28.6 | 24.0 | 25.5 | 30.4 |
| JRDB-Base [6] | 21.5 | 44.9 | 27.7 | 54.3 | 45.9 | 48.5 | 38.4 | 33.1 | 34.8 | 37.0 |
| PAR [12] | 54.3 | 44.2 | 46.9 | 50.3 | 52.5 | 50.1 | 42.1 | 24.5 | 30.3 | 42.4 |
| MUP [2] | 56.8 | 45.6 | 48.6 | 55.7 | 49.7 | 51.3 | 57.0 | 46.2 | 47.3 | 49.2 |
| THSM (Ours) | **59.6** | **48.4** | **50.7** | **58.2** | **54.1** | **54.7** | **60.1** | **47.9** | **52.0** | **52.5** |

**Table 3: Comparison results of different methods under conventional multi-person activity recognition setting.**

| Methods | Individual Action | | | Global Activity | | |
|---|---|---|---|---|---|---|
| | $\mathcal{P}_i$ | $\mathcal{R}_i$ | $\mathcal{F}_i$ | $\mathcal{P}_p$ | $\mathcal{R}_p$ | $\mathcal{F}_p$ |
| AT [10] | 36.8 | 30.1 | 31.7 | 17.4 | 15.7 | 16.1 |
| SACRF [25] | 39.2 | 29.4 | 32.2 | 34.8 | 26.2 | 28.4 |
| TCE+STBiP [47] | 46.6 | 37.7 | 39.7 | 31.9 | 23.7 | 26.4 |
| HiGCIN [45] | 36.9 | 30.1 | 31.6 | 46.0 | 34.2 | 38.0 |
| PAR [12] | 51.0 | 40.5 | 43.4 | 52.8 | 31.8 | 38.8 |
| MUP [2] | 55.4 | 44.8 | 47.7 | 58.0 | 49.0 | 51.8 |
| THSM (Ours) | **58.2** | **47.3** | **50.1** | **66.3** | **53.6** | **57.8** |

self-attention layers are implemented in a multi-head fashion, where the number of heads is 8. The dimension of the features is set to 512, i.e., $D = 512$. The temperature factor $\tau^\ell$ in Eq. (5) is empirically set to 0.2. To train the proposed hierarchical framework, we adopt the ADAM optimizer [17], with a learning rate of $1 \times 10^{-4}$ and a weight decay of $5 \times 10^{-4}$, for 60 epochs. The batch size is set to 8. As argued in [12], integrating temporal clues requires additional multi-object associations and group evolution detection, which has the risk of introducing unexpected errors, especially in challenging panoramic crowd scenes. Thus, following previous works [2, 12], the temporal information across frames is not taken into consideration.

## 4.2 Comparison with State-of-the-Arts

**Results in Clustered Group Division Setting.** For a comprehensive performance comparison, in addition to existing panoramic activity recognition models, e.g., PAR [12] and MUP [2], we also include several state-of-the-art social group-activity understanding methods, e.g., ARG [41], SA-GAT [5], and JRDB-Base [6], which have been modified for adaptation to the target task. Table 1 tabulates the comparison results of our proposed THSM method with other

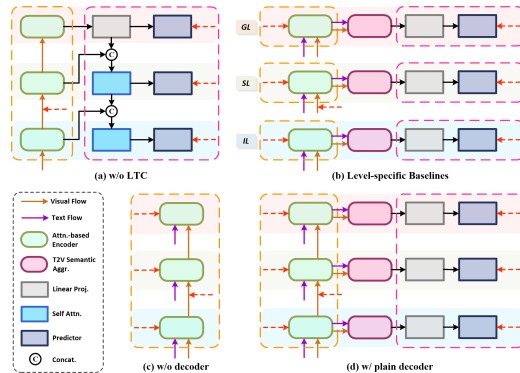

**Figure 3: Structure comparison of four types of baseline models.**

state-of-the-art methods based on clustered group division setting. Following the practice of [12], a spectral clustering algorithm [49] was applied to the individual-relation matrix produced by ARG [41] for group division, and a feature fusion mechanism [6] was further employed to conduct complete panoramic activity recognition. As can be seen from Table 1, the proposed THSM framework achieves consistent performance improvements over other competing methods in all evaluation metrics by considerable margins. Compared with the pioneering hierarchical GNNs-based model, i.e., PAR [12], the proposed method leads to an overall F1-score improvement of 9.5%. Moreover, our proposed THSM framework improves the state-of-the-art MUP [2] by 3.6% in terms of $\mathcal{F}_a$ metric.

**Results in Ground-truth Group Division Setting.** We include some more state-of-the-art methods, i.e., AT [10], SACRF [25], TCE+STBiP [47], and HiGCIN [45], which are originally developed based on the traditional group activity

**Table 4: Ablation results of the proposed THSM framework with and without using label-text clues.**

| Ablation Config. | Individual $\mathcal{F}_i$ | Group $\mathcal{F}_p$ | Global $\mathcal{F}_g$ | Overall $\mathcal{F}_a$ |
|---|---|---|---|---|
| w/o LTC | 44.5 | 24.6 | 42.2 | 37.1 |
| Full THSM | 50.1 | 27.3 | 57.8 | 45.1 |

**Table 5: Ablation results of using hierarchical modeling and separate level-specific modeling strategies.**

| Ablation Config. | Individual $\mathcal{F}_i$ | Group $\mathcal{F}_p$ | Global $\mathcal{F}_g$ |
|---|---|---|---|
| IL | 43.1 | - | - |
| SL | - | 24.8 | - |
| GL | - | - | 39.6 |
| IL+ SL+ GL | 50.1 | 27.3 | 57.8 |

recognition pipeline, for comparison. However, these methods can neither detect latent social groups in crowd scenes nor generate the individual-relation matrix as in ARG. Thus, following [12], the ground-truth group detection results are provided as additional input for performance evaluation. As summarized in Table 2, all the listed methods exhibit significant performance gains (at least 18% in terms of $\mathcal{F}_p$ metric) in social group activity recognition, which indicates the influence of precise group division in the downstream task. More importantly, we can find that the proposed THSM method consistently outperforms other competitors in all evaluation metrics.

**Generalization Evaluation on Conventional Multi-person Activity Recognition.** We evaluate the generalization ability of different methods on conventional multi-person activity recognition, which consists of two subtasks, i.e., individual action and global activity recognition. Table 3 presents the comparison results by adopting the original setting of previous group activity recognition methods without modification. It can be observed that our proposed method achieves the best F1 scores of 50.1% and 57.8% in recognizing individual action and global activity, respectively.

## 4.3 Ablation Studies

**Effect of the Injection of Label-text Clues.** Since the proposed method learns hierarchical action/activity semantics with the aid of label texts, we conducted ablation experiments to investigate its effect. For comparison, as shown in Fig. 3(a), we implemented a baseline model, without exploiting textual clues of labels. Specifically, it takes only the visual embeddings as the input of the hierarchical encoder-decoder framework and is trained by optimizing the losses defined in Eqs. (10)-(11). Table 4 tabulates the ablation results. We can find that the baseline model "w/o LTC" still slightly outperforms the hierarchical GNNs-based model, i.e., PAR [12], by 1.5%, in terms of the overall F1 score. By injecting label-text embeddings, the full version of the proposed THSM framework significantly improves the $\mathcal{F}_a$ score of the baseline model by 8%, which shows the advantage of establishing multi-level cross-modal semantics associations

**Table 6: Ablation results of using different alignment loss functions in the encoder part.**

| Alignment Losses | | | Individual | Group | Global | Overall |
|---|---|---|---|---|---|---|
| $\mathcal{L}_{enc}^{\mathcal{I}}$ | $\mathcal{L}_{enc}^{\mathcal{S}}$ | $\mathcal{L}_{enc}^{\mathcal{G}}$ | $\mathcal{F}_i$ | $\mathcal{F}_p$ | $\mathcal{F}_g$ | $\mathcal{F}_a$ |
| - | - | - | 46.2 | 25.0 | 51.4 | 40.9 |
| - | - | ✓ | 46.5 | 25.1 | 53.2 | 41.6 |
| - | ✓ | - | 47.1 | 25.9 | 52.0 | 41.7 |
| ✓ | - | - | 48.3 | 25.5 | 52.6 | 42.1 |
| - | ✓ | ✓ | 46.9 | 26.3 | 54.8 | 42.7 |
| ✓ | - | ✓ | 47.7 | 26.0 | 55.4 | 43.0 |
| ✓ | ✓ | - | 48.7 | 26.8 | 56.1 | 43.5 |
| ✓ | ✓ | ✓ | 50.1 | 27.3 | 57.8 | 45.1 |

**Table 7: Ablation results under different settings of the decoder.**

| Ablation Config. | Individual $\mathcal{F}_i$ | Group $\mathcal{F}_p$ | Global $\mathcal{F}_g$ | Overall $\mathcal{F}_a$ |
|---|---|---|---|---|
| w/o decoder | 45.3 | 24.3 | 42.7 | 37.4 |
| w/ plain decoder | 46.5 | 25.1 | 55.6 | 42.4 |
| w/ C2F decoder | 50.1 | 27.3 | 57.8 | 45.1 |

between the label space and visual content in recognizing multi-grained activities.

**Effect of the Hierarchical Modeling.** To study the effect of the hierarchical modeling shown in Fig. 3(b), we implemented three level-specific baseline models, which are separately tailored for recognizing individual action, social group activity, and global activity, respectively. Concretely, each baseline model is constructed based on a single-level encoder-decoder architecture and is fed with the visual and label-text embeddings at the corresponding level. The ablation results are presented in Table 5. Without cross-level semantics interactions, the separate level-specific baseline models exhibit an obvious performance drop of 7%, 2.5%, and 18.2%, in terms of F1 scores, in the three sub-tasks of panoramic activity recognition, respectively. In contrast, the hierarchical modeling strategy employed in our proposed THSM method simultaneously tackles the three sub-tasks in a unified framework, which can facilitate the flow of useful knowledge across different semantic granularities.

**Effect of the Alignment Loss.** We conduct ablation experiments to investigate the influence of visual-textual semantic alignment losses, which are imposed on different levels of the encoder part. Table 6 presents the ablation results. We can find that removing the alignment loss at a specific level will lead to degraded performance in recognizing the corresponding activities. What's worse, it also has a side effect on activity recognition on other semantic levels. For instance, without using the individual-level alignment loss (the fifth row in Table 6), i.e., $\mathcal{L}_{enc}^{\mathcal{I}}$, the model shows a performance drop of 3.2%, in terms of $\mathcal{F}_i$, in individual action recognition, and also degrades the $\mathcal{F}_p$ and $\mathcal{F}_g$ metrics by 1% and 3%, in recognizing social group and global activities, respectively. The performance of the proposed THSM framework can be improved when continually introducing the cross-modal alignment losses on different semantic levels.

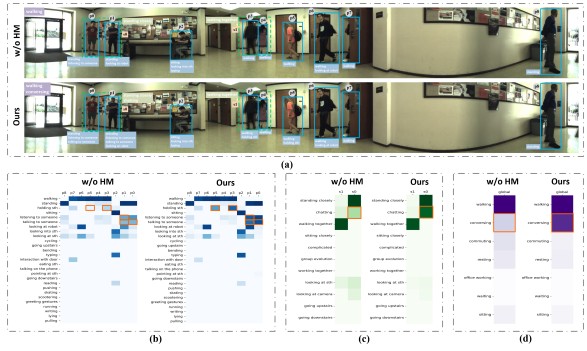

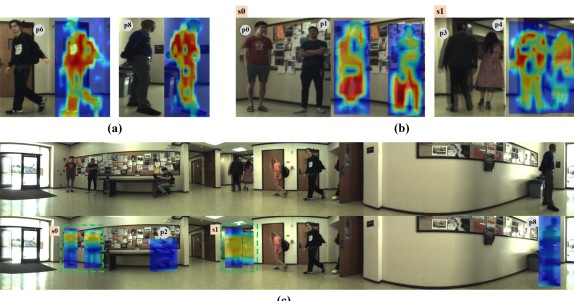

Figure 4: Visualization of (a) panoramic activity recognition results and the learned (b) individual-level, (c) social group-level, and (d) global-level visual-to-textual semantic affinity matrices, produced by the baseline model trained without using hierarchical modeling (w/o HM) and the proposed THSM framework (ours).

Figure 5: Visualization of attention maps of the proposed THSM framework, activated by (a) individual-level, (b) social group-level, and (c) global-level action/activity categories.

**Effect of the Coarse-to-Fine Decoder.** To evaluate the contribution of the coarse-to-fine (C2F) decoder, we implemented two baseline models for comparison. The vanilla baseline, denoted as "w/o decoder" (see Fig. 3(c)), is built by solely maintaining the hierarchical encoder, which directly produces the action/activity recognition results from the semantic affinity matrix $\mathbf{A}^\ell$ at each semantic level. Moreover, another baseline model, i.e., "w/ plain decoder" (see Fig. 3(d)), takes the learned cross-modal representation $\mathbf{g}^\ell$ after text-to-visual aggregation, as the input of a two-layer MLP-based decoder, for inferring actions/activities at different semantic granularities. As illustrated in Table 7, "w/ plain decoder" outperforms "w/o decoder" by 5%, in terms of $\mathcal{F}_a$, which suggests that even a simple encoder can be augmented by the text-to-visual semantic aggregation. In addition, by progressively integrating higher-level contexts with lower-level features, the proposed C2F decoder can further lead to an improvement of 2.7%, in terms of the overall F1 score.

## 4.4 Qualitative Results

**Visualization of Learned Visual-to-Textual Semantic Affinities.** To intuitively interpret the learned cross-modal semantic affinities, as shown in Fig. 4, we visualize the three-level visual-to-textual affinity matrices (i.e., $\mathbf{A}^{\mathcal{I}}$, $\mathbf{A}^{\mathcal{S}}$, and $\mathbf{A}^{\mathcal{G}}$ derived from Eq. (5)) and the corresponding panoramic activity recognition results of the proposed THSM framework and the three level-specific baseline models. For the individual-action level, due to the lack of holistic guidance from other levels, the baseline model fails to capture the affinities between subtle appearance cues with the semantics conveyed by the label-text embeddings (highlighted by orange bounding boxes in Fig. 4(b)), e.g., the imperceptible talking behaviors for person "0" and "1", and the unobservable bottle held by person "3" owing to small size and motion blur. This eventually results in missing some

action categories. For the social group-activity level, without ingesting sufficient fine-grained atomic action clues, the level-specific baseline model can solely explicitly establish semantic relations between group "0" and the text of "standing closely" (see Fig. 4(c)), which leads to the missing of "chatting". Similarly, without cross-level interactions, the global-level baseline model only identifies the salient event semantics and assigns a relatively low affinity score to the label-text of "conversing" (see Fig. 4(d)), thus casing incomplete activity recognition results.

**Visualization of Attention on the Tokens.** To qualitatively examine the effectiveness of the proposed THSM method, as shown in Fig. 5, we visualize the attention maps of the tokens, activated by the actions or activities at different semantic granularities. In Fig. 5(a), we can find that the proposed method assigns relatively higher weights on the tokens regarding the crucial body parts, e.g., legs and eyes, for person "6". This helps to accurately recognize the individual actions of "walking" and "looking at robot (camera)". In Fig. 5(c), for the holistic scene, our proposed THSM framework pays less attention to the irrelevant individuals, e.g., persons "2" and "8", and highlights more on the groups "0" and "1", which can provide useful cues for recognizing the global activities of "conversing" and "walking".

## 5 CONCLUSION

In this paper, we propose a label text-aided hierarchical semantics mining (THSM) method, which targets at explicitly exploring multi-granularity cross-modal associations for improving panoramic activity recognition (PAR). Concretely, the proposed THSM framework is designed based on a three-level encoder-decoder architecture. The encoder establishes the hierarchical correspondence between visual content and label texts from multiple semantic levels, while the decoder progressively integrates the higher-level contextual knowledge into the lower-level cross-modal representations for coarse-to-fine action/activity recognition. Both quantitative and qualitative evaluation results on the public JRDB-PAR data set demonstrate the superior performance of the proposed method.

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
