# OpenReview forum: "Label Text-aided Hierarchical Semantics Mining for Panoramic Activity Recognition"
_acmmm.org/ACMMM/2024/Conference — MM2024 Poster_

### Official Review · Reviewer_Lum5 · 2024-05-20

**Rating:** 5
**Confidence:** 3

**Summary:**

The paper addresses the challenging task of understanding crowd scenes by identifying multi-grained human behaviors, including individual actions, social group activities, and global activities. The authors propose a novel label text-aided hierarchical semantics mining (THSM) framework that aims to capture cross-modal associations between visual content and label texts, thereby addressing the limitations of previous studies that solely relied on video input.

**Strengths:**

The introduction of a label text-aided hierarchical semantics mining framework is a significant contribution to the field. The hierarchical encoder and decoder designed to align and integrate semantic representations at different granularities demonstrate a well-thought-out approach to addressing the complexity of panoramic activity recognition.
The use of both a hierarchical encoder and decoder to progressively integrate lower-level representations with higher-level contextual knowledge for coarse-to-fine recognition is methodologically sound and innovative.
The extensive experimental results on the public JRDB-PAR benchmark provide strong evidence for the effectiveness of the proposed framework. The superior performance over state-of-the-art methods is a compelling aspect of the study.

**Limitations:**

While the paper outlines the general structure and components of the THSM framework, more detailed explanations and clarifications of certain methodological aspects could enhance the comprehensibility and reproducibility of the work. For instance, detailed descriptions of the hierarchical encoder and decoder architectures, as well as the training process, would be beneficial.
Although the experimental results show the superiority of the proposed framework, a more in-depth comparative analysis with existing methods, including a discussion of specific scenarios where THSM outperforms others, would strengthen the argument for its effectiveness.

**Suitability:**

2

---

### Official Review · Reviewer_z1Qg · 2024-05-25

**Rating:** 3
**Confidence:** 3

**Summary:**

Previous research has often focused on extracting cross-granularity activity-semantics relationships solely from the video data, neglecting the inherent semantic hierarchy present within the label-text domain. In response, this paper introduces a framework known as Text-Aided Hierarchical Semantic Mining (THSM). This framework delves into multi-level cross-modal relationships by establishing a hierarchical semantic alignment between the visual content and the associated label texts.

**Strengths:**

1. The paper proposes a label text-aided hierarchical semantics mining (THSM) framework, which explores multi-level cross-modal associations by learning hierarchical semantic alignment between visual content and label texts, while it seems like the idea of many traditional multi-level alignment frameworks, such as [1,2,3].

[1] Image search with text feedback by visiolinguistic attention learning. CVPR 2020
[2] Multi-modal transformer with global-local alignment for composed query image retrieval. TMM 2023
[3] MUP: Multi-granularity Unified Perception for Panoramic Activity Recognition. MM 2023

2. The experiments conducted are sufficient, which show the learned multi-level semantics.

**Limitations:**

1. The technical novelty is limited to me. The main contribution proposed by the author is hierarchical semantic mining. However, this technology of using different levels of semantic information to improve the performance of the model is outdated and has been widely used many years ago.

2. The fonts in the figures designed by the author are too small, such as Figures 1, 3 and 4. The author needs to adjust the size to improve readability. Also, there seems to be something wrong with the template and it needs to be checked carefully.

**Suitability:**

3

---

### Official Review · Reviewer_92eZ · 2024-06-06

**Rating:** 4
**Confidence:** 2

**Summary:**

These work presents a label text-aided hierarchical semantics mining (THSM) framework for panoramic activity recognition (PAR) in crowded scenes. The main contributions of the paper include the proposal of the THSM framework, which leverages hierarchical cross-modal semantic correspondence between visual content and label texts to improve PAR. Additionally, a multi-level encoder-decoder architecture is introduced, aiming to align visual-textual semantics at different granularities and integrate cross-level cross-modal semantics for coarse-to-fine action/activity recognition. The paper also provides extensive experimental results and ablation studies on the public JRDB-PAR benchmark, demonstrating the superiority of the THSM framework over existing methods.

The novel aspects of this work include its exploration of hierarchical cross-modal semantic correspondence for PAR and the multi-level encoder-decoder architecture, distinguishing it from existing research in activity recognition and addressing the challenges of recognizing multi-grained human behaviors in crowded scenes. The contributions significantly advance panoramic activity recognition by introducing the THSM framework and demonstrating its effectiveness. This framework provides valuable insights for recognizing individual actions, social group activities, and global activities in crowded scenes, with the potential to influence future research and development in the field.

**Strengths:**

* The proposed label text-aided hierarchical semantics mining (THSM) method introduces a novel approach to explicit exploration of multi-granularity cross-modal associations for improving panoramic activity recognition (PAR).

* The THSM framework is designed based on a three-level encoder-decoder architecture, establishing hierarchical correspondence between visual content and label texts from multiple semantic levels, demonstrating a robust theoretical foundation.

* The document demonstrates technical correctness through extensive experimental results and ablation studies, showcasing the superior performance of the proposed THSM framework over state-of-the-art methods in clustered group division, ground-truth group division, and conventional multi-person activity recognition settings.

* The comprehensive evaluation results, including comparisons with existing methods and visualization of attention maps, provide a clear understanding of the effectiveness of the proposed THSM framework.

* The document is well-structured and clearly presents the proposed THSM framework, experimental results, and ablation studies, ensuring clarity in the presentation of the research findings.

**Limitations:**

It appears that the authors have not released the code, which compromises the reproducibility of the study. There are some typos present in the manuscript, as well as incomplete figures, such as fig.3. Additionally, the paper references a limited number of recent works from 2023/2024, which may reduce its relevance when compared to the latest research.

**Suitability:**

3

---

### Official Review · Reviewer_ax5Y · 2024-06-10

**Rating:** 3
**Confidence:** 3

**Summary:**

This work aims to enhance panoramic activity recognition by mining the hierarchical semantic relationships among individual, group, and global levels. Specifically, when obtaining the multi-modal representation of one level, it leverages information from lower levels. For fine-grained predictions, it uses information from coarse-grained levels. The order from coarse-grained to fine-grained levels is global-level, group-level, and individual-level. State-of-the-art results were achieved on a recently released dataset JRDB-PAR.

**Strengths:**

1. **Motivation:** The motivation behind this paper is quite clear.
2. **Writing:** The paper is well-organized.

**Limitations:**

1. **Method:** About the proposed method, there are a few minor points that are unclear:
    a. Why use the outdated text embedding model GloVe instead of BERT?
    b. How to get the $M \times M$ matrix $\tilde{Z}$ in Group Detection Loss?

**Suitability:**

3

---

### Meta-Review · Area_Chair_q2xL · 2024-07-02

**Recommendation:** Accept (Poster)
**Confidence:** 5

**Metareview:**

This paper receives one weak accept, one borderline accept, and two borderline rejects initially. The reviewers raise some questions regarding text embedding, group detection loss, technical novelty, more in-depth comparative analysis with existing methods, etc. Most of these questions are addressed in the rebuttal and recognized by reviewers, with the minor concern remaining in technical novelty. Eventually with the merit of the work, this paper receives one borderline accept, one borderline reject, one borderline accept (initial score without final update), one weak accept (initial score without final update). Considering the novelty justification in the rebuttal, the AC recommended to accept. Authors are encouraged to revise the paper according to the reviews, including adding the novelty justification into the revised paper.